# High-Temperature Corrosion of Flame-Sprayed Power Boiler Components with Nickel Alloy Powders

**DOI:** 10.3390/ma16041658

**Published:** 2023-02-16

**Authors:** Artur Czupryński, Janusz Adamiec, Marcin Adamiak, Marcin Żuk, Antonin Kříž, Claudio Mele, Monika Kciuk

**Affiliations:** 1Department of Welding, Silesian University of Technology, Konarskiego Street 18A, 44-100 Gliwice, Poland; 2Department of Metallurgy and Recycling, Silesian University of Technology, Krasińskiego Street 8, 40-019 Katowice, Poland; 3Materials Research Laboratory, Silesian University of Technology, Konarskiego Street 18A, 44-100 Gliwice, Poland; 4Faculty of Mechanical Engineering, Institute of Technology and Materials, Jan Evangelista Purkyně University in Ústí nad Labem, Pasteurova Street 1, 400 96 Ústí nad Labem, Czech Republic; 5Department of Engineering for Innovation, University of Salento, via per Monteroni, 73100 Lecce, Italy; 6Department of Engineering Materials and Biomaterials, Silesian University of Technology, Konarskiego Street 18A, 44-100 Gliwice, Poland

**Keywords:** flame powder spray process, coating, nickel alloy, high temperature corrosion, sheet pile walls, power boiler

## Abstract

The development trends in the energy sector clearly indicate an increase in the share of biomass and alternative fuels fed for combustion in power boilers, which results in the imposition of many unfavourable factors and a demanding working environment. During the operation of the energy system, this means a sharp increase in corrosion of the gas-tight pipe walls and coils by the destructive action of chlorine and sulphur. Implementing advanced surface protection in addition to the selection of materials of better quality and resistance to difficult working conditions would significantly reduce their wear by high temperature corrosion. Thermally sprayed coatings offer a great opportunity to protect machine components and energy systems against corrosion, erosion, impact load and abrasive wear. This article presents the test results of high-temperature corrosion resistance of coatings made with Ni-Cr-B-Si and Ni-B-Si alloy powders on a boiler steel substrate. Samples with sprayed coatings were exposed to an atmosphere with a composition of N_2_ + 9% O_2_ + 0.08% SO_2_ + 0.15% HCl at 800 °C for 250, 500, 750 and 1000 h. Tests results of coatings made of Ni-Cr-B-Si alloys subjected to the influence of a corrosive environment showed the formation of a layer of scale on the surface, composed mainly of Cr_2_O_3_ oxide, which was a passive layer, reducing the rate of corrosion. Coatings sprayed with Ni-B-Si alloys showed significantly lower corrosion resistance. It was found that the developed technology of subsonic flame spraying with powders of the Ni-Cr-B-Si type allows the production of coatings compliant with the requirements of the energy industry, which allows their use as anti-corrosion protection on boiler elements intended for waste disposal and biomass combustion.

## 1. Introduction

High costs of energy produced from energy resources (natural gas, crude oil, fossil coals) and the need to protect the natural environment more and more often encourage world economies to use renewable energy sources obtained from wind, sunlight or water. Biomass is a natural energy resource, which, due to its origin, is divided into: wood biomass from trees and shrubs; green biomass from plants with non-lignified stems and whose above-ground part dies after the cultivation period; fruit biomass from plant parts containing seeds; and a mixture resulting from the deliberate mixing of biofuels [1]. The increase in the share of biomass and alternative fuels fed for combustion in power boilers is associated with surface phenomena that may accompany the contact of waste gases with boiler elements. Corrosion processes in waste incineration installations are mainly caused by the presence of moisture (surface corrosion, crevice corrosion, pitting corrosion, stress corrosion, intercrystalline corrosion) and high operating temperature (sulfiding, carbonization, dusting, nitriding corrosion, sulfiding). In the combustion chamber of a power boiler, corrosion processes may be of the following nature:−oxidizing due to the presence of nitric and chromic acids, iron and copper salts, nitrites, chromates and urea; −reducing due to the presence of sulphuric, hydrochloric and phosphoric acids, organic acids, a solution of alkaline salts and halides;−mixed (oxidizing-reducing) related to the presence of a mixture of the above-mentioned compounds. 

High-temperature corrosion, also known as low NOx corrosion, occurs most often in boilers with high steam parameters and mainly attacks the boiler structure elements heated to high temperatures. It is a chemical process of metal oxidation at high temperatures, exhaust gas atmospheres or environments containing sulphur or hydrogen sulphide [2,3]. The specificity of high-temperature corrosion consists in the fact that a homogenous, passive layer of oxides does not form on the surface of the metal, as is the case under normal oxidation conditions. In addition to the correct chemical composition of the exhaust gas, high temperature is required for this type of corrosion to occur.

According to Ostrowski et al. [4], with oxygen deficiency in the furnace of a power boiler, sulphur, chlorine and fluorine react with oxygen, forming a protective layer of oxides on the surface of the panel elements of the gas-tight pipe wall and coils. These elements then react with the iron, causing a rapid loss of wall thickness of these elements.

High-temperature corrosion is manifested by the destruction of the material and the loss of stress properties. This often causes failures related to leaks in pipes and coils and, consequently, the need to replace them, which leads to costs related to boiler downtime [5,6]. Resistance to aggressively corrosive environments is ensured by the modification of the structure and properties of construction materials and the use of protective coatings with the appropriate chemical composition of the alloy. In addition to high mechanical properties, these materials must also show heat resistance, creep resistance and thermal shock resistance, and be characterized by low susceptibility to corrosion at high temperatures and aggressive atmospheres containing, among others, sulphuric compounds, oxygen or water vapour [7]. The main group of materials resistant to high-temperature corrosion are nickel-based alloys. These alloys are characterized by very high wear resistance when used in particularly difficult working conditions. The characteristics of this group of materials are: the ability to work at temperatures up to 1250 °C; limited sensitivity to variable and dynamic loads; and resistance to highly aggressive corrosive environments of gases, sulphur compounds, nitrogen and carbon [8]. Incoloy 625, 686 and 825 superalloys are a popular group of nickel alloys. In these alloys, as in high-alloy austenitic stainless steels, additions of alloying elements such as: Mo, Si, Cu, Ti, Nb and low C content improve creep strength, as well as corrosion resistance and resistance to acids and alkalis [9,10,11]. According to Rozmus-Górnikowska et al. [12], it is related to the formation of passive oxide layers containing Cr_2_O_3_ and Fe_3_O_4_ on the surface of materials. With the increase in the chromium content in nickel alloys, the parabolic course of oxidation is extended due to changes in the oxide morphology, starting from Fe_2_O_3_/Fe_3_O_4_ through Fe_3_O_4_/(Fe,Cr)_3_O_4_ and Fe_3_O_4_/(Fe,Cr)_3_O_4_/Cr_2_O_3_ to the layer of pure Cr_2_O_3_, Figure 1c. The constant change of oxide morphology contributes to better protection of surfaces subjected to corrosion [7,13].

On the other hand, uninterrupted operation of the power unit is conducive to the development of corrosive processes of material destruction. Corrosion products are characterized by a layered structure with various properties, such as roughness, porosity or adhesion to the substrate. As a result, the oxide layer periodically peels off, which, with constant temperature jumps, affects the oxidation kinetics. Consequently, these processes lead to a thinning of the wall thickness and an increase in stress in the material of the gas-tight pipe wall and coils. The mechanism of the build-up of oxide layers on the surface of pipes is a beneficial phenomenon, as it inhibits further progress of corrosion phenomena. Nevertheless, as a result of the increase in the thickness of the oxide layer, heat exchange deteriorates, which causes an increase in the wall temperature, and thus promotes the degradation of the material structure [13,14].

Most often, the following methods are used to produce protective layers on gas-tight wall pipes and coils of power boilers: MIG (metal inert gas), PPTA (plasma powder transferred arc) or HPDDL (high power direct diode laser) [15,16,17]. However, these technologies do not always meet the high requirements related to the lack of welding imperfections, minimum depth of the heat affected zone and low iron content in the made cladding layer. Other methods of protecting boiler components include: −thermal spraying of ceramic coatings, e.g., Al_2_O_3_, Cr_2_O_3_, TiO_2_, ZrO_2_ [18,19];−use of composite pipes, e.g., 3R12/4L7 (304L/P265GH), Sanicro 38/HT8 (Incoloy 825/10CrMo9-10) and Sanicro 63/HT7 (Incoloy 625/X10CrMoVNb9-1) [20,21,22]. 

The use of thermally sprayed ceramic coatings is associated with many technical and economic challenges, such as the large dimensions of the power unit or the need to use a recirculation system to cool the boiler. This solution turned out to be technically complex and very expensive at the stage of implementation and operation. Currently, this type of anti-corrosion protection has been abandoned and can only be found in older power units. The structures of newly built power boilers are based on gas-tight walls and coils made of composite pipes or pipes protected with hardfacing layers [4,20,23]. 

Composite pipes with an outer layer of austenitic stainless steel, e.g., 3R12/4L7, did not fully meet the hopes placed in them, because various cracks or flaking of the entire layer often occurred in the cladding layer. According to the tests presented in [5,24], the cracks were caused by the difference between the coefficient of thermal expansion for austenitic steel and carbon structural steel, which in the temperature range of 30–500 °C is respectively, α_ss_ = 18.4 × 10^−6^ 1/K and α_cs_ = 14.3 × 10^−6^ 1/K, as well as thermal fatigue and, stress corrosion cracking. Sanicro 38/4L7 grade composite pipes with Incoloy 825 (α_inc_ = 16.2 × 10^−6^ 1/K) superalloy outer layer turned out to be only slightly better than 3R12/4L7 grade composite pipes with 306L austenitic steel layer. In this case, cracks were also found in the plating and at the interface between the two layers [5]. On the other hand, composite pipes of the Sanicro 63/4L7 grade with an outer layer of Incoloy 625 (α_inc_ = 15.3 × 10^−6^ 1/K) superalloy are characterized by very good properties, but they are not commonly used due to the high cost of production and the impossibility of making high-quality pipes of required lengths [5,13,24].

Welding technologies of applying layers and coatings resistant to corrosion, erosion and high temperatures, implemented on an industrial scale, still require comprehensive research, contributing to the production of high-quality protection of boiler elements that meet the increasingly demanding requirements of the power industry. Problems caused by periodic failures of power boilers that are already in operation require effective service actions using uncomplicated technologies. The high-temperature corrosion resistance tests undertaken were aimed at determining the resistance to high temperature and aggressive exhaust gas environment of coatings flame-sprayed with self-fluxing powders made of Ni-Cr-B-Si alloys and non-fluxing powders made of Ni-B-Si alloys. These alloys are considered to be easily weldable, and their properties are comparable to Incoloy superalloys [9,10,11,12,13,25]. The addition of boron causes the formation of eutectic with a lower melting point than pure metals, which allows the coating to be melted without melting the substrate. The silicon contained in the powder binds all impurities in the form of oxides that float to the surface during melting. In addition, these alloys are strengthened by the precipitation of carbides such as Ni_6_C, Cr_23_C_6_ and B_4_C. 

The main aim of the conducted research was to confirm the ability of developed technology of subsonic flame spraying with powders of the Ni-Cr-B-Si and Ni-B-Si type to the production of coatings compliant with the requirements of standards and expectations of the energy industry, as anti-corrosion protection on boiler elements intended for waste disposal and biomass combustion.

## 2. Materials and Research Methods

The flame powder spraying process was carried out on a production stand equipped with a modern and precise, modular oxy-acetylene system for manual flame powder spraying CastoDyn DS 8000 (Castolin, Gliwice, Poland), Figure 2b. Thermally sprayed coatings were made on a fragment of a gas-tight pipe wall of a powder boiler consisting of a pipe ϕ 51.0 × 6.3 mm welded to flat bars 6.0 mm thick, Figure 1b. 

All structural elements were made of low-alloy 16Mo3 (1.5415) boiler steel. Before the spraying process, the surface of the samples, in accordance with the requirements of ISO 2063-1 [26], was subjected to abrasive blasting in a cabin sandblaster, using an abrasive medium in the form of broken cast iron shot with G34 granulation (grain size 0.8–1.2 mm). This procedure was aimed at cleaning the external surface of the samples from impurities such as rust, scale or grease. After this stage of processing, the samples were additionally brushed to remove any traces of grit, and then chemically cleaned with tetrachloroethylene. The final surface roughness of the substrate after shot-blasting was Ra = 12 μm, Rz = 85 μm. As an additional material for flame spraying, two types of self-fluxing metallic powders made of Ni-Cr-B-Si alloys (Eutalloy^®^ RW 17535 and Metco^®^ 15E) were used and, for comparison, two types of non-fluxing metallic powders of the Ni-B-Si type (Eutalloy^®^ BronzoChrom10185 and Eutalloy^®^ NiTec 10224). Chemical composition and physical properties of low-alloy boiler steel according to EN 10273 [27] and powders for spraying according to manufacturers’ data are given in Table 1, Table 2, Table 3, Table 4 and Table 5.

Each sample was preheated to 150 °C immediately before the spraying process. The burner was run in a wall-mounted position (PC) covering the entire surface of the protected element, and then the direction of spraying was changed several times by 90 degrees until a coating with a thickness of about 600 µm was obtained. After the spraying process, the coatings made with self-fluxing powders from remeltable alloys of the Ni-Cr-B-Si type were melted by heating with an oxy-acetylene flame of a gas burner to a temperature in the liquidus–solidus range of the alloy used. Coatings sprayed with Ni-B-Si alloys have not been remelted. Spray parameters are shown in Table 6.

Visual tests (VT) and penetration tests (PT) of coatings sprayed with self-fluxing powders made of Ni-Cr-B-Si alloys and powders made of Ni-B-Si non-fluxing alloys, carried out in accordance with the requirements of ISO 17637:2017 [28] and ISO 3452:2021 [29], did not show any non-conformity of coating surface and shape. No cracks, porosity or lack of adhesion to the substrate were found in the coatings. The established parameters of thermal spraying ensured the production of coatings with an acceptable level of quality that met the requirements of class “B” according to ISO 5817:2014 [30]. On the basis of preliminary tests, the influence of spraying parameters and the type of metal powder used on the depth of the heat affected zone (HAZ) “h”, the degree of porosity of the coating “p” and the roughness of the coating surface “r” were determined. In addition, the following parameters were also determined: the thickness of the sprayed coating “g” and the iron content on the surface of the sprayed coating “Fe”, Table 7.

The assessment of the surface microgeometry of the thermally sprayed coatings was performed using a surface roughness gauge SRT-200 (Sunpoc, Guiyang City, China). Surface roughness measurements were made on the length of the elementary section of 2.5 mm. Microscopic examinations were carried out on standardly prepared metallographic specimens. The samples to be observed with a light microscope were etched in two stages: the native material of the steel pipe was revealed using FeCl_3_Et (Mi19Fe) solution, and the structure of the nickel alloy coatings was revealed by electrochemical etching with the following reagent: 20 cm^3^ HCl, 10 g FeCl_3_, 30 cm^3^ of distilled water. Observation and recording of microstructure images were carried out using an Olympus SZX7 stereoscopic microscope (Olympus Corporation, Tokyo, Japan) and an Olympus GX 71 inverted metallographic microscope (Olympus Corporation, Tokyo, Japan). The porosity of the surface area fraction of the sprayed coatings was measured by computer image analysis using the MicroScan computer program for Windows. 

The prerequisite for the recognition of the technology of corrosion protection of pipes used in the construction of a heat exchanger operating in biomass incineration plants is the assessment of the iron content on the surface of the thermally sprayed coating. In their recommendations, certification bodies require that the iron content in the surface layer does not exceed 7% [31]. The XRF X-MET8000 Expert portable spectrometer (Hitachi High-Technologies Corporation, Tokyo, Japan), commonly used in industrial conditions, was used to analyse the iron content on the surface of coatings sprayed with nickel alloys selected for testing. The tests were carried out at 10 points spaced 10 mm apart, Figure 2d. Sample results of microscopic examinations are shown in Figure 3, and all measurements of characteristic sizes are summarized in Table 7.

High-temperature corrosion resistance tests of the sprayed coatings were carried out on samples with an area of 250 mm^2^ (25 mm × 10 mm) cut from the central part of the flat bar of the spectacle joint of a fragment of the gas-tight pipe wall, Figure 2b. Native material (steel 16Mo3) was removed from the samples taken before the corrosion test. The assessment of corrosion resistance concerned only coatings sprayed with metallic powders from Ni-Cr-B-Si and Ni-B-Si alloys. For comparative purposes, tests of high-temperature corrosion resistance of 16Mo3 steel (1.5415) were also carried out independently, from which the elements of the gas-tight pipe wall of the power boiler were made.

The diagram of the test stand consisting of tube furnaces with built-in quartz and ceramic tubes, reducers, gas intake points with rotameters, a neutralizing system and cylinders with gases included in the simulated gas mixture is shown in Figure 4. 

The tested samples were placed in Al_2_O_3_ refractory crucibles and closed in the test chamber of the tube furnace. A mixture of gases with the chemical composition: N_2_ + 9.0% O_2_ + 0.08% SO_2_ + 0.15% HCl was used as the working atmosphere. The chemical composition of the gas mixture was similar to the atmosphere present in power boilers. The tests were carried out at a temperature of 800 °C. The experiment procedure was divided into three stages:−heating the furnace charge in an argon atmosphere up to the set temperature (800 °C) in order to avoid the oxidation of samples;−withstanding of the samples in a corrosive atmosphere with the given concentrations of gases at a given temperature and the flow of the gas mixture ensuring a single exchange of the atmosphere within 4 h;−cooling of the furnace charge in an argon atmosphere, up to 20 °C.

Corrosion resistance tests were carried out for up to 1000 h, with measurements of the mass increase of corrosion products every 250 h. Three weight measurements were made for each of the samples, which were averaged. The percentage relative standard deviation (RSD%) for samples with corrosion protective coatings was ≤1%. The change in mass (Δm) of the sample after the tests was adopted as a measure of the resistance to high-temperature corrosion of the coatings. The samples were weighed on an AS 60/220.R2 PLUS analytical balance (Radwag, Radom, Poland) with an accuracy of 0.0001 g. 

Studies of the corrosion mechanisms of thermally sprayed coatings after corrosion tests were carried out in a scanning electron microscope Zeiss Supra 25 (Carl Zeiss AG, Oberkochen, Germany) using the EDS method—energy dispersive spectrometry. 

X-ray diffraction tests enabling the phase analysis of coatings were performed using a Panalytical X’Pert Pro MPD diffractometer (Malvern Panalytical Ltd., Malvern, UK), using filtered radiation (Kβ Fe filter) of a lamp with a cobalt anode (λKα = 0.179 nm). The diffractograms were recorded in the Bragg—Brentano geometry, using a PIXcell 3D detector on the axis of the diffracted beam, in the range of angles 20–110 [2θ] (step = 0.05°, counting time per step = 100 s). The obtained diffraction patterns were analysed using the dedicated Panalytical High Score Plus software together with the PAN-ICSD structural database. The tests of chemical and phase composition made it possible to define the types of corrosion products formed after the corrosion tests.

Electrochemical tests on corrosion resistance were also performed on all tested coatings. Electrochemical study of corrosion resistance was conducted in accordance with the standard ISO 17475:2010 [32]. Corrosion resistance of the material was examined by Atlas 0531 EU (Atlas-Sollich ZSE, Rębiechowo, Poland) unit potentiostat/galvanostat and results were generated by the AtlasCorr05 software. The base steel for corrosion examinations was cut into 10 mm × 10 mm plates. Before starting the corrosion tests, specimens were cleaned in ethanol and dried in air. All electrochemical tests were performed in 3.5% NaCl solution, which was freshly prepared prior to the testing. 

A typical three-electrode system was employed in this test set-up where the Ag/AgCl electrode was used as the reference electrode, the tested steel was the anode (0.384 cm^2^ exposed area) and stainless steel was the counter electrode for current measurements. The potential was changed in the anodic direction at the rate of 1 m·V·s^−1^. All the electrochemical tests were performed at a constant room temperature.

## 3. Results and Discussion

### 3.1. Flame Spraying with Nickel-Based Powders

Technological tests of anti-corrosion protection on a fragment of a gas-tight pipe wall were carried out by thermal spraying with the use of self-fluxing powders made of remeltable Ni-Cr-B-Si alloys and powders of non-fluxing Ni-B-Si alloys. These types of tubular walls are used for the production of heat exchangers in waste incineration boilers. Thermally sprayed coatings with nickel-based alloys significantly increase the resistance of pipes to corrosion and erosion [33,34,35]. They can be manufactured on new tubular structures or in the event of the need to repair elements of a panel damaged during operation. In many studies, the authors pointed to technological difficulties in the process of surfacing and thermal spraying with nickel-based alloys. Most often, they resulted from incorrectly selected padding parameters, which may cause an increase in the Fe content in the pad layer above the required value of 7% [12,31] and unacceptable quality and adhesion of thermally sprayed coatings.

The Fe content in the overly layer is exponentially dependent on the amount of heat supplied to the base material. During arc surfacing, the requirement to obtain the desired value below 7% Fe in the overlay requires limiting the line energy of the surfacing to approx. 3 kJ/cm [15,17]. The increase in the value of the linear energy of the surfacing also affects the increase in the thickness of the nickel oxide (NiO) layer, the presence of which on the surface of the gas-tight pipe wall is unacceptable. After the surfacing process, these oxides must be removed from the surface of the pipe wall. In the case of thermally sprayed coatings, the critical parameters are proper surface preparation before spraying and ensuring the diffusive nature of the coating-substrate connection.

In the proposed technology of flame powder spraying of coatings with a thickness of about 600 μm, such difficulties do not occur. No external inconsistencies were found in all the coatings made. The average value of the Ra roughness parameter for coatings made of Ni-Cr-B-Si alloys ranged from 2.03 μm to 2.48 μm, and for coatings made of Ni-B-Si alloys it ranged from 10.93 μm to 11.54 μm. These results are consistent with the data presented in [34,35]. A significant internal inconsistency of the coatings made of Ni-B-Si alloys is the porosity of the coatings, which reached 9.6%. According to Czupryński et al. [35], it is possible to flame spray powders made of Ni-B-Si non-meltable alloy with porosity even below 6%. In the case of coatings flame sprayed with self-fluxing powders made of remeltable Ni-Cr-B-Si alloys, the degree of porosity of the coatings can be significantly reduced by remelting the coating in the liquidus–solidus temperature range of the alloy used [34]. 

The obtained test results confirmed that it is possible to produce coatings from Ni-Cr-B-Si alloys with a porosity of 1.7%, Table 7. The coatings were characterized by proper adhesion, indicating the mechanical-diffusion nature of the connection with the substrate. The smallest amount of intermediate phases in the coating was observed in the case of coatings made of Ni-Cr-B-Si alloys. Intermediate phases were located in the metal deposit at the line of connection of the coating with the base material. In each case, a clearly separated heat affected zone (HAZ) could be observed in the native material. A maximum HAZ depth of more than 105 μm was recorded for a coating sprayed with Eutalloy^®^ RW 17535 powder (Ni-Cr-B-Si) and a minimum HAZ depth of approximately 77 μm for a coating sprayed with Eutalloy^®^ BronzoChrom 10185 powder (Ni-B-Si). The increase in HAZ depth in samples with coatings of remeltable Ni-Cr-B-Si alloys resulted from the additional heat supplied to the base material in the remelting process. The absence of the base material in the coating provides better control of the Fe content in the metal deposit. The highest average Fe content = 3.7% was determined on the surface of the coating sprayed with Metco^®^ 15 remeltable alloy powder. The iron content on the surface of the other coatings was less than 1%, Table 7.

It was found that the factors determining the quality of coatings flame-sprayed with powders based on nickel alloys on the surface of elements of a gas-tight pipe wall are:−proper preparation and cleaning of the substrate surface before spraying in order to obtain good adhesion of the coating,−limiting the amount of heat supplied to the base material by maintaining the proper distance between the burner and the sprayed surface and the use of a slightly carburizing flame (λ = 1.2), ensuring low content of Fe and NiO oxides on the surface of the coating,−the use of remelting the sprayed coating by heating it with an oxy-acetylene flame of a gas burner to a temperature in the liquidus–solidus range of the alloy used in order to minimize the internal porosity of the metal deposit and obtain the appropriate smoothness of the coating surface.

It was also shown that there is no need for forced internal cooling of the heat exchanger tube in order to limit the depth of the diffusion zone and the heat affected zone in the base material and to ensure the appropriate quality of the sprayed coatings.

### 3.2. Evaluation of High-Temperature Corrosion Resistance of Flame-Sprayed Coatings Made of Ni-Cr-B-Si and Ni-B-Si Powders

The results of testing the high-temperature corrosion resistance of coatings sprayed with flame-sprayed powders made of Ni-Cr-B-Si and Ni-B-Si alloys selected for testing, as well as the construction material from which the elements of the gas-tight pipe wall were made, i.e., 16Mo3 (1.5415) steel, are presented in Table 8. The course of changes in the mass of samples with anti-corrosion coatings is also illustrated in Figure 5 and Figure 6. The curves in the graph show changes in the mass of the sample per unit area as a function of the duration of the corrosion test.

The conducted analysis of the course of the corrosion process, based on the empirical relationship between the scale mass increase or metal loss and the time of the oxidation reaction, indicates the parabolic nature of the sample mass changes during the course of the experiment. Flame-sprayed coatings with self-fluxing powders made of remeltable alloys Ni-Cr-B-Si of the Eutalloy^®^ RW 17535 and Metco^®^ 15E grades showed a negligible tendency to corrosive wear, Figure 5. The corrosion rate of these coatings under the conditions of the experiment was 0.0016 ((mg/cm^2^)/h) and 0.0018 ((mg/cm^2^)/h), Figure 7. These results indicate that the Fe content in the coating sprayed with Metco^®^ 15E powder of about 4% does not cause the formation of a large amount of iron oxide Fe_2_O_3_ on the surface of the coating, which has much lower adhesion to the substrate than chromium oxide Cr_2_O_3_, which in the conditions of corrosive and erosive wear could intensify the processes of coating degradation. Coatings flame-sprayed with powders made of Ni-B-Si non-meltable alloys of Eutalloy^®^ BronzoChrom 10185 and Eutalloy^®^ NiTec 10224 alloys, containing about 0.5% Fe, performed slightly worse in terms of corrosive wear. The corrosion rate of these coatings in given temperature-time conditions was, respectively, 0.0033 (mg/cm^2^)/h) and 0.0030 (mg/cm^2^)/h), Figure 7. However, in the case of these coatings, it can be seen that the mass increase of corrosion products is the most increased in the time interval of 250–500 h. After this period, the corrosion processes progressed more slowly. In addition, it was observed that after 500 h of exposure to high temperature and corrosive atmosphere, the wear resistance of coatings made of non-meltable alloys decreased, which was manifested by strong overheating of the coating structure. On the other hand, the unprotected construction material (16Mo3 steel) was characterized by the highest corrosion rate, which after 1000 h of the test was 0.1715 ((mg/cm^2^)/h), Figure 6. 

The general results of susceptibility to high-temperature corrosion for the adopted temperature-time conditions indicate a higher corrosion resistance of coatings made of Ni-Cr-B-Si alloys than coatings of Ni-Si-B alloys, which was slightly affected by the Fe content up to 4%, Figure 5 and Figure 7.

The high-temperature corrosion resistance tests were supplemented by metallographic tests. Exemplary surfaces of samples after corrosion tests and the surface of the joint observed in the scanning electron microscope are shown in Figure 8. The tests were carried out in the technique of recording secondary electrons (SE) and backscattered secondary electrons (BSE). The SE images show the surface topography, while the BSE images inform about the varied chemical composition. In further studies, the corrosion products were identified by microanalysis of the EDS chemical composition (Figure 9) and the identification of the phase composition by XRD (Figure 10).

Microscopic examinations on cross-sections of corroded samples after 1000 h showed that in the case of a 16Mo3 steel sample, the thickness of the oxide layer, mainly Fe_3_O_4_, reaches even 1.25 mm.

On the basis of SEM microscopic observations and tests of the chemical composition of the surface of coatings sprayed with self-fluxing powders made of Ni-Cr-B-Si alloys after 1000 h of corrosion tests, the presence of places with high Cr content was found, Figure 9a,b. The analysis of the phase composition showed that the main corrosion product of all alloy sprayed samples was Cr_2_O_3_ chromium oxide. On the surface of the sample containing about 4% wt. of Fe, sites with a relatively high iron content were found, which would indicate the presence of this element oxides, Figure 9b. This was confirmed by the study of the phase composition, Figure 10. On the other hand, the high content of silicon in the layer of corrosion products reaching even more than 15% wt. is presumably related to the furnace lining. 

The kinetics and mechanism of nickel oxidation are the subject of numerous studies due to the formation of NiO oxide on the surface of this metal, which, due to its transport properties, is treated as a model semiconductor. Despite the huge number of works devoted to this subject, there is still no full agreement in the literature regarding both the mechanism of high-temperature corrosion of nickel and the structure of nickel oxide defects. Some authors assume that in the process of oxide scale growth on nickel at temperatures lower than 1000 °C, apart from the core network diffusion of cations, the core diffusion of oxygen across grain boundaries is involved [36]. At higher temperatures, it is assumed that the core diffusion of the metal plays a decisive role, but there is no agreement as to the degree of ionization of defects [37]. 

In the case of both coatings flame sprayed with Ni-B-Si non-meltable alloy powders, after 1000 h of corrosion tests carried out at 800 °C and in a highly corrosive gas atmosphere N_2_ + 9.0% O_2_ + 0.08% SO_2_ + 0.15% HCl, the oxidation process can be described with the kinetic parabolic law. The oxide scale formed on the surface of the coating was mainly composed of the NiO phase, Figure 10c,d. Nickel oxide is a p-type semiconductor with a metal deficiency, so its growth was due to core diffusion of Ni ions. The incorporation into the NiO lattice of cations with a valency higher than the valence of nickel ions causes an increase in the vacancy concentration in the cation sublattice, and thus the rate of oxidation increases. The presence of monovalent metals improves the protective properties of the scale, e.g., the presence of metals such as chromium or manganese increases the rate of nickel oxidation. According to Rivoaland et al. [36], each alloy addition has a certain concentration limit value at which, under certain reaction conditions, a saturated MeO–NiO solid solution is formed, and then the scale is two- or multi-phase. Most often, however, there is a two-layer scale, which consists of an outer layer made of the NiO phase and an inner layer consisting of a mixture of both MeO + NiO alloy components. If the alloying element has a greater affinity with oxygen than nickel, then a protective, single-phase scale layer composed only of the alloying element oxide may form on the surface of the alloy. Such additives may be aluminium, chromium and silicon.

In addition, nickel borides Ni_3_B, which are characterized by high hardness and thermal resistance, were identified on the surface of the coatings. The melting point of Ni_3_B nickel borides is 1478 °C, and their hardness is in the range of 10.49–10.75 GPa. According to González et al. [38], the presence of boron in the nickel coating reduces the diffusion movement of atoms and vacancies in the crystal lattice, increases the resistance to plastic deformation and reduces the size of crystallites during electro-crystallization and their growth during recrystallization, and also strengthens the nickel matrix.

On the surface of 16Mo3 grade steel, the results of the analysis of chemical and phase composition confirmed the presence of mainly Fe_3_O_4_ iron oxide with a small amount of Cr_2_O_3_ chromium oxide.

Examination of corrosion process products produced on coatings sprayed with self-fluxing powders from remeltable Ni-Cr-B-Si alloys showed their laminar morphology. On the surface of coatings sprayed with Eutalloy^®^ RW 17535 and Metco^®^ 15E powders, a Cr_2_O_3_ oxide film is formed, which is a barrier to oxygen access and significantly reduces the intensity of corrosion. The presence of Fe in sprayed coatings can lead to the formation of Fe_3_O_4_ oxide, the adhesion of which is much lower. In the iron oxide layer, unfavourable cracks are observed, which cause flaking of the layer, and thus a decrease in the corrosion resistance of the coating, Figure 8e,f. Therefore, limiting the Fe content in powders for flame spraying of coatings protecting against high-temperature corrosion in the range of 0.5–4% is reasonable, although the results of corrosion kinetics studies would indicate the possibility of increasing the Fe content up to about 7%. 

The analysis of the obtained results indicates a high corrosion resistance of coatings sprayed with self-fluxing powders made of Ni-Cr-B-Si alloys in a gas atmosphere simulating flue gases in coal-fired boilers. A uniform layer of corrosion products was observed on the surface of all samples. This layer causes a slight reduction in heat exchange between the flue gases and the 16Mo3 steel sample material, at the same time it also protects well against further corrosion progress. The results of the microanalysis of the chemical composition of gas corrosion products indicate that they are rich in chromium, nickel and oxygen, which probably indicates the presence of a layer of chromium oxide Cr_2_O_3_ on the surface of the samples. Its presence protects samples against further oxidation, ensuring adequate corrosion resistance at high temperatures. This is advantageous due to the fact that the Cr_2_O_3_ layer is continuous and adheres well to the coating, and under high-temperature corrosion conditions it quickly rebuilds, which ensures permanent corrosion protection. The analysis of changes in the weight of the samples during the experiment also confirms that a Cr_2_O_3_ film is formed on the surface, which covers the surface and hinders the access of the corrosive atmosphere to the surface. This is indicated by the initial increase in the mass of the samples (for 750 h) and then the slowdown of the growth (up to 1000 h), which confirms the formation of a continuous and tight layer of chromium oxide (Figure 5). 

The results of potentiodynamic experiments conducted in water solution 3.5% NaCl are shown in Table 9 and the course of changes of current density as a function of potential are presented in Figure 11. 

In each case, a characteristic course of the polarization curve is observed. It was found that the analyzed samples had varied resistance to the corrosion. 

In terms of the polarization resistance Rp and corrosion current density Icor, Eutalloy^®^ RW 17535 seems to have a higher degree of corrosion resistance than other tested alloys due to the high content of chromium, which determines good corrosion resistance in the aggressive environment of chloride ions. However, Metco^®^ 15E alloy display also excellent corrosion resistance properties (I_cor_ = 674 × 10^−9^ A/cm^2^, R_p_ = 41.91 kΩ × cm^2^). 

Alloys Eutalloy^®^ NiTec 10224 and Eutalloy^®^ BronzoChrom 10185 showed similar the lowest corrosion resistance (I_cor_ = 2.586 × 10^−6^ A/cm^2^, R_p_ = 6.06 kΩ × cm^2^ and I_cor_ = 3.285 × 10^−6^ A/cm^2^, R_p_ = 5.39 kΩ × cm^2^ respectivly).

The tests have shown that use of flame-sprayed coatings with self-fluxing powders made of remeltable Ni-Cr-B-Si alloys on the surfaces of gas-tight pipe wall panel elements and coils can effectively protect them against high-temperature corrosion and extend the lifetime of the power installation several times.

Compared to previously used surfacing technologies and the main advantage of flame spraying with self-fluxing powders from meltable alloys is the high quality of the coatings and the lack of substrate material in the metal deposit. In addition, this technology makes it possible to spray complex shapes even in hard-to-reach places. Further research will be devoted to the comparison of high temperature corrosion resistance of coatings sprayed with FPS and HVOF powders made of Ni-Cr-B-Si alloys.

## 4. Conclusions

Based on the conducted research and analysis of the results, the following conclusions were formulated:During flame powder spraying of boiler elements with nickel alloys, the amount of heat supplied to the base material should be limited by maintaining the proper distance between the burner and the sprayed surface and using a slightly carburizing flame (λ = 1.2), ensuring low Fe content on the coating surface and avoiding the need to the use of internal cooling of the element to limit overheating of the base material.Coatings flame sprayed with self-fluxing powders from remeltable Ni-Cr-B-Si alloys and non-fluxing powders from Ni-B-Si alloys are resistant to high-temperature corrosion in the exhaust gas atmosphere (N_2_ + 9.0% O_2_ + 0.08% SO_2_ + 0.15% HCl) typical for power boiler installations. At temperatures up to 800 °C, high-temperature corrosion in the range from 0 to 1000 h runs in a parabolic manner. Coatings made of Ni-Cr-B-Si alloys show about 50% higher resistance to high-temperature corrosion than coatings made of Ni-B-Si alloys and much lower susceptibility to corrosion than low-alloy steel intended for boiler structures. The corrosion rate of coatings made of Ni-Cr-B-Si alloys in the conditions of the experiment was 0.0030–0.0033 (mg/cm^2^)/h and was 52–57 times lower than the corrosion rate of low-alloy 16Mo3 (1.5415) boiler steel.The results of potentiodynamic tests conducted in an aqueous solution of 3.5% NaCl confirmed the high corrosion resistance of coatings sprayed with self-fluxing powders from remeltable Ni-Cr-B-Si alloys.The main corrosion product of coatings made of remeltable alloys Ni-Cr-B-Si is Cr_2_O_3_ oxide, which forms a thin passivation layer on the surface of the coating, reducing the rate of corrosion. On the surface of coatings made of non-smelting alloys Ni-B-Si, NiO_2_ and NiB_3_ phases are formed, which protect the surface less effectively against the aggressive exhaust gas environment and create an unfavourable thermal barrier limiting heat transfer. In the case of 16Mo3 boiler steel, a thicker layer of oxides is formed on its surface, mainly Fe_3_O_4_, which cracks and does not provide anti-corrosion protection.

## Figures and Tables

**Figure 1 materials-16-01658-f001:**
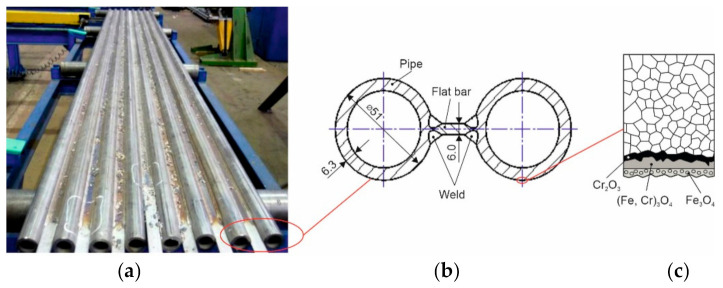
View of the gas-tight wall of a power boiler: (**a**) view of the panel before the surface protection process, (**b**) spectacle joint, (**c**) diagram of the formation of oxides on the surface of fine-grained austenitic steel at elevated temperature.

**Figure 2 materials-16-01658-f002:**
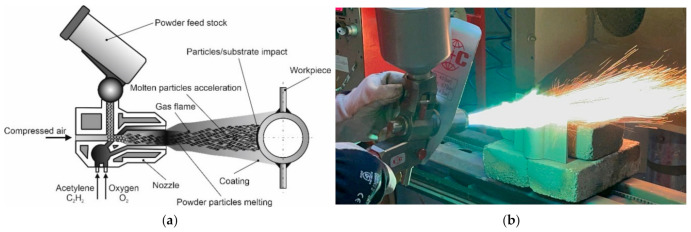
Flame powder spraying process: (**a**) diagram of a manual flame powder spray burner, (**b**) view of a test of flame powder spraying of a remeltable Ni-Cr-B-Si alloy coating using a CastoDyn DS 8000 burner, (**c**) view of the remelting test of the coating sprayed with a self-fluxing powder made of the remeltable Ni-Cr-B-Si alloy, (**d**) view of the sample with the coating protecting against high-temperature corrosion.

**Figure 3 materials-16-01658-f003:**
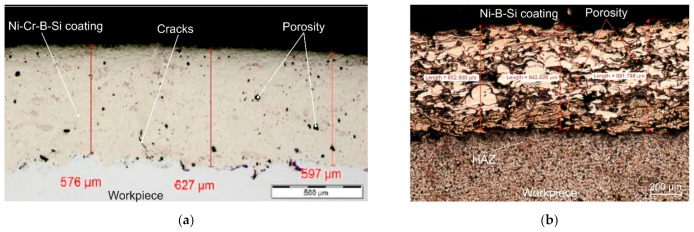
View of the microstructure of the cross-section of coatings sprayed with self-fluxing powder made of Ni-Cr-B-Si alloy (Eutalloy^®^ RW 17535) and powder made of Ni-B-Si non-fluxing alloy (Eutalloy^®^ BronzoChrom 10185) and base material of low-alloy boiler steel of the 16Mo3 grade: (**a**) microstructure of the Ni-Cr-B-Si alloy shell, (**b**) microstructure of the Ni-B-Si alloy shell, (**c**) image of porosity of the inner shell of Ni-Cr-B-Si alloy, (**d**) image of porosity of the inner shell made of Ni-B-Si alloy, (**e**) image of HAZ microstructure and base material under Ni-Cr-B-Si alloy coating, (**f**) image of HAZ microstructure and base material under Ni-B-Si alloy coating.

**Figure 4 materials-16-01658-f004:**
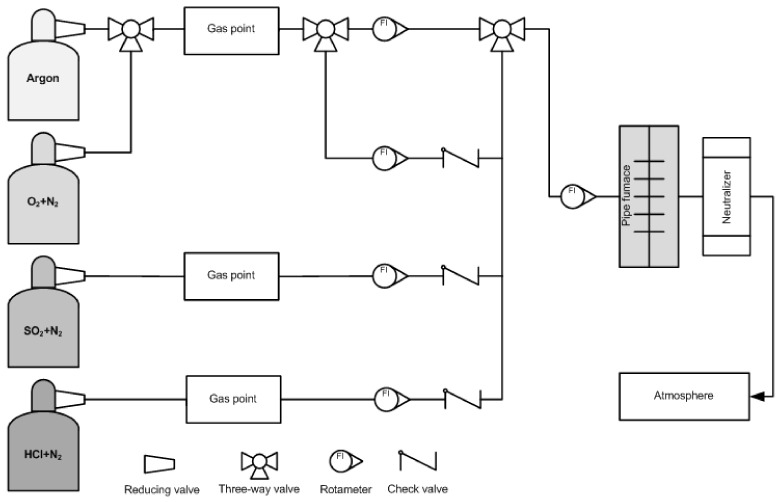
Scheme of equipment for corrosion tests.

**Figure 5 materials-16-01658-f005:**
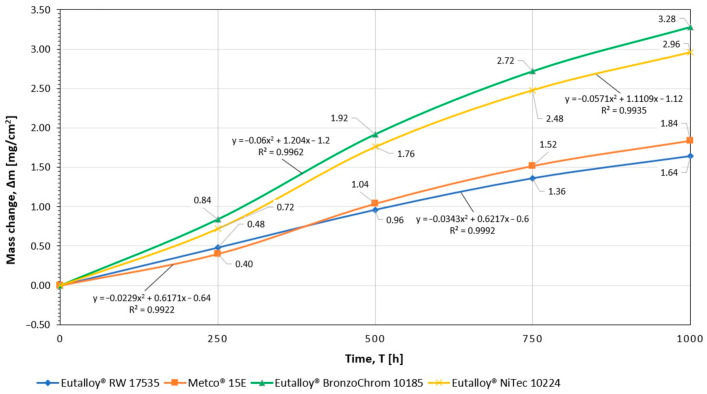
Graph of the unit mass change of samples protected with protective coatings as a function of exposure to a corrosive atmosphere.

**Figure 6 materials-16-01658-f006:**
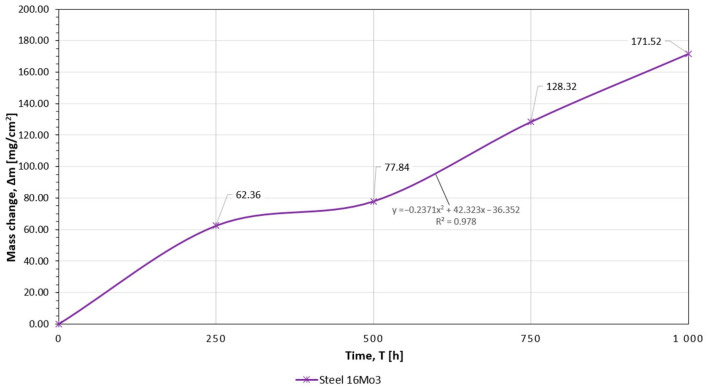
Graph of the unit weight change of 16Mo3 (1.5415) steel base material samples as a function of exposure to corrosive atmosphere.

**Figure 7 materials-16-01658-f007:**
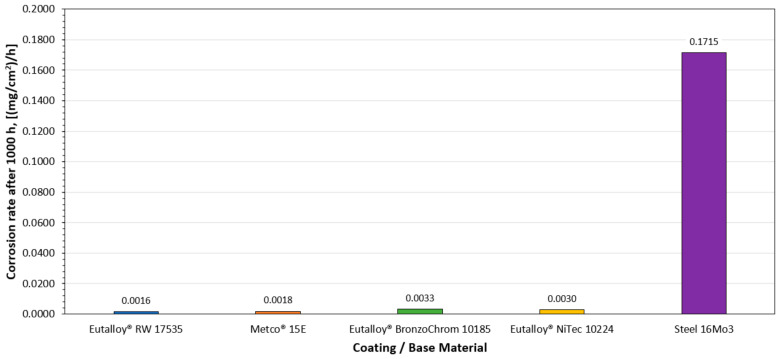
Graph of corrosion rate of protective coatings and base material exposed to corrosive atmosphere at 800 °C for 1000 h.

**Figure 8 materials-16-01658-f008:**
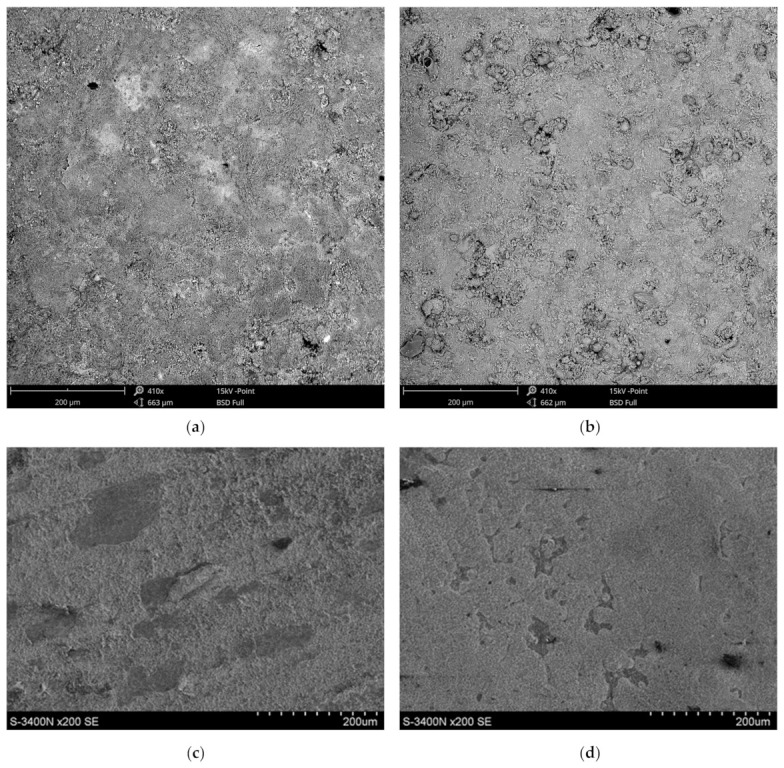
Surface morphology after high-temperature corrosion resistance tests at 800 °C for 1000 h: (**a**) flame-sprayed coating with Eutalloy^®^ RW 17535 powder, (**b**) flame-sprayed coating with Metco^®^ 15E powder, (**c**) flame-sprayed coating with Eutalloy^®^ BronzoChrom 10185, (**d**) flame-sprayed coating with 10185 Eutalloy^®^ NiTec 10224 powder, (**e**) cracks in flame-sprayed coating with Eutalloy^®^ RW 17535 powder, (**f**) cracks in flame-sprayed coating with Metco^®^ 15E powder.

**Figure 9 materials-16-01658-f009:**
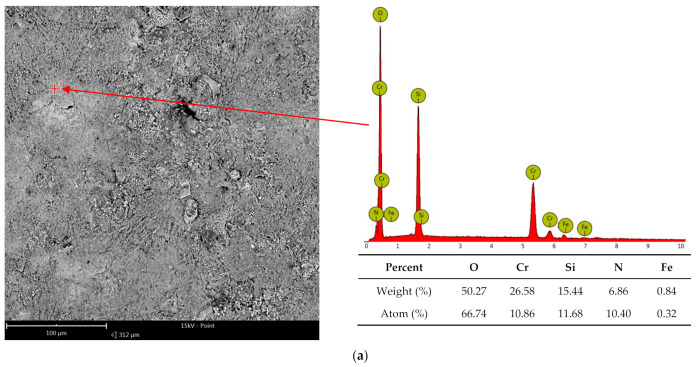
Results of microanalysis of the chemical composition of corrosion products after high-temperature corrosion resistance tests at 800 °C for 1000 h of flame-sprayed coatings: (**a**) Eutalloy^®^ RW 17535, (**b**) Metco^®^ 15E, (**c**) Eutalloy^®^ BronzoChrom 10185, (**d**) Eutalloy^®^ NiTec 10224.

**Figure 10 materials-16-01658-f010:**
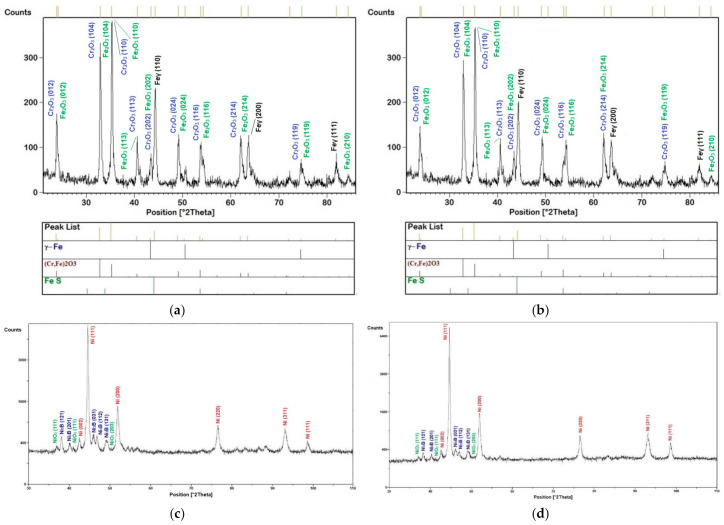
Results of analysis of the phase composition of corrosion products after high-temperature corrosion resistance tests at 800 °C for 1000 h of flame-sprayed coatings: (**a**) Eutalloy^®^ RW 17535, (**b**) Metco^®^ 15E, (**c**) Eutalloy^®^ BronzoChrom 10185, (**d**) Eutalloy^®^ NiTec 10224.

**Figure 11 materials-16-01658-f011:**
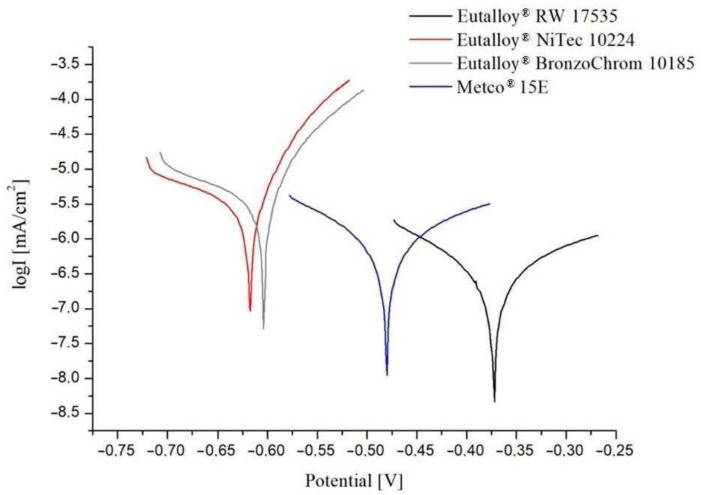
Potentiodynamic polarization curves of analyzed specimens.

**Table 1 materials-16-01658-t001:** Chemical composition and physical properties of 16Mo3 (1.5415) low-alloy boiler steel according to EN 10273.

Chemical Composition, wt.%										
C	Mn	Si	P	S	Cr	Mo	Ni	Cu	N	Fe
0.12–0.20	0.4–0.9	<0.35	<0.025	<0.010	<0.3	0.25–0.35	<0.3	<0.3	<0.012	rest
Mechanical properties										
Strength R_m_, MPa		Hardness, HB 30			Melting range (liquidus/solidus), °C			Thermal resistance, °C		
440–590		130–170			1460/1420			530		

**Table 2 materials-16-01658-t002:** Chemical composition of Ni-Cr-B-Si metal deposit according to the manufacturer’s data.

Chemical Composition, wt.%					
C	Cr	Fe	B	Si	Ni
Eutalloy^®^ RW 17535 (Castolin)					
0.8	26	1.0	3.0	3.7	rest
Metco^®^ 15E (Sulzer)					
1.0	17	4.0	3.5	4.0	rest

**Table 3 materials-16-01658-t003:** Physical properties of the Ni-Cr-B-Si metal deposit according to the manufacturer’s data.

Mechanical Properties of the Coating Weld Metal ^(1)^		
Hardness, HV30	Density, g/cm^3^	Thermal resistance, °C
Eutalloy^®^ RW 17535 (Castolin)		
480	7.8-8.0	≤870
Metco^®^ 15E (Sulzer)		
745	7.8–8.1	≤840

Note: ^(1)^ alloy in the initial state, not heat-treated, test temperature 20 °C.

**Table 4 materials-16-01658-t004:** Chemical composition of the Ni-B-Si metal deposit according to the manufacturer’s data.

Chemical Composition, wt.%					
C	Cr	Fe	B	Si	Ni
Eutalloy^®^ BronzoChrom 10185 (Castolin)					
≤0.1	≤0.5	≤0.5	2.5	3.0	rest
Eutalloy^®^ NiTec 10224 (Castolin)					
≤0.1	≤0.5	≤0.5	1.5	2.0	rest

**Table 5 materials-16-01658-t005:** Physical properties of the Ni-B-Si metal deposit according to the manufacturer’s data.

Mechanical Properties of the Coating Metal Deposit ^(1)^			
Hardness, HV30	Density, g/cm^3^	Melting range (liquidus/solidus), °C	Thermal resistance, °C
Eutalloy^®^ BronzoChrom 10185 (Castolin)			
390	8.9	1050/1175	≤600
Eutalloy^®^ NiTec 10224 (Castolin)			
240	8.1	1050/1280	≤600

Note: ^(1)^ alloy in the initial state, not heat treated, test temperature 20 °C.

**Table 6 materials-16-01658-t006:** Parameters of flame-sprayed Ni-Cr-B-Si and Ni-B-Si coatings using CastoDyn DS 8000 torch.

Sample Number	Type of Powder	Oxygen Pressure [bar]	Acetylene Pressure [bar]	Air Pressure [bar]	Number of the Orifice for the Powder
1	Eutalloy^®^ RW 17535	4	0.7	1	4
2	Metco^®^ 15E	4	0.7	1	4
3	Eutalloy^®^ BronzoChrom 10185	4	0.7	2	4
4	Eutalloy^®^ NiTec 10224	4	0.7	2	4

Note: Standard modular nozzles regulating the flame outlet (SSM 20) were used. Slightly carburizing flame was used. The spraying distance was 200 mm.

**Table 7 materials-16-01658-t007:** Geometric and metallurgical properties of sprayed coatings.

Sample Number	Type of Coating	Geometrical Properties		Metallurgical Properties		
		g, µm	Ra, μm	h, µm	p, %	Fe, wt.%
1	Eutalloy^®^ RW 17535	600	2.03	95	1.7	0.9
2	Metco^®^ 15E	628	2.48	131	2.6	3.7
3	Eutalloy^®^ BronzoChrom 10185	661	11.54	83	9.6	0.4
4	Eutalloy^®^ NiTec 10224	634	10.93	74	8.3	0.3

Note: Average values are given in the table.

**Table 8 materials-16-01658-t008:** Change in the weight of samples protected with protective coatings and the base material as a function of exposure to a corrosive atmosphere at a temperature of 800 °C.

Time, T [h]	Mass Change Δm, [mg]														
	Eutalloy^®^ RW 17535			Metco^®^ 15E			Eutalloy^®^ BronzoChrom 10185			Eutalloy^®^ NiTec 10224			Steel 16Mo3 (1.5415)		
	$\bar{X}$	S(x)	RSD%	$\bar{X}$	S(x)	RSD%	$\bar{X}$	S(x)	RSD%	$\bar{X}$	S(x)	RSD%	$\bar{X}$	S(x)	RSD%
0	0	0	0	0	0	0	0	0	0	0	0	0	0	0	0
250	0.012	0.17	0.29	0.010	0.20	0.33	0.021	0.30	0.50	0.018	0.20	0.33	1.559	0.40	0.66
500	0.024	0.20	0.33	0.026	0.10	0.17	0.048	0.30	0.51	0.044	0.30	0.50	1.946	0.50	0.83
750	0.034	0.35	0.58	0.038	0.56	0.93	0.068	0.40	0.66	0.062	0.40	0.66	3.208	0.70	1.17
1000	0.041	0.30	0.50	0.046	0.30	0.50	0.082	0.60	1.00	0.074	0.50	0.83	4.288	1.2	2.0

**Table 9 materials-16-01658-t009:** Results of potentiodynamic tests for examined specimens.

Material	I_cor_, [A/cm^2^]	E_cor_, [mV]	R_p_, [kΩ × cm^2^]
Eutalloy^®^ RW 17535	283 × 10^−9^	−376	110
Metco^®^ 15E	674 × 10^−9^	−481	41.91
Eutalloy^®^ BronzoChrom 10185	3.285 × 10^−6^	−607	5.39
Eutalloy^®^ NiTec 10224	2.586 × 10^−6^	−619	6.06

## Data Availability

Not applicable.

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
