# Peer review of "High-Temperature Corrosion of Flame-Sprayed Power Boiler Components with Nickel Alloy Powders"

_materials, 2023, doi:10.3390/ma16041658_

Round 1
Reviewer 1 Report
see attach

Author Response
Dear Reviewer 1
Please see the attachment.
Yours sincerely,
Artur Czupryński

Reviewer 2 Report
Authors have studied studied corrosion behaviour of flame sprayed nickel alloy components for boilers application. The language of the paper is clear and the work is presented well. However, I have concerns about the novelty of the work. A lot of similar work is presented by many researchers, some of them are as follows:
Sang, Kezheng, and Yugang Li. "Cavitation erosion of flame spray weld coating of nickel-base alloy powder." Wear 189.1-2 (1995): 20-24.
Kezheng, Sang, and Li Yugang. "Cavitation erosion of flame spray weld coating of nickel-base alloy powder." International Journal of Multiphase Flow 22.S1 (1996): 153-153.
Croopnick, Gerald A. "Nickel based thermal spray powder and coating, and method for making the same." U.S. Patent No. 10,240,238. 26 Mar. 2019.
Sturgeon, Andrew. "The corrosion behavior of hvof sprayed stainless steel and nickel alloy coatings in artificial seawater." CORROSION 2003. OnePetro, 2003.
Menasri, N., Zergane, S., Aimeur, N., & Saci, A. Experimental Investigation on the Coating of Nickel-Base Super Alloy Using Wire Flame Spraying. Acta Universitatis Sapientiae, Electrical and Mechanical Engineering, 14(1), 28-39.
Sidhu, T. S., S. Prakash, and R. D. Agrawal. "Hot corrosion studies of HVOF sprayed Cr3C2–NiCr and Ni–20Cr coatings on nickel-based superalloy at 900 C." Surface and Coatings Technology 201.3-4 (2006): 792-800.
Authors must justify/mention the novelty of the present work at the end of the introduction section.
I have following comments to improve the quality of the manuscript:
1. Abstract section is too long. General description of the problem should be removed and included in the introduction section.
2. Some self-citations seem to be irrelevant. Either they should be removed or they should be explained in a better way so that they citation is justifiable.
3. Experiments are conducted with only four samples by varying only gas pressure. More parameters can be varied to have a comprehensive study.
4. Fig. 3 (a): the label craks should be cracks
5. Conclusions are obvious and in line with previous studies. What is the key contribution of this study, must be highlighted.
Author Response
Dear Reviewer 2
Please see the attachment.
Yours sincerely,
Artur Czupryński

Reviewer 3 Report
This work focused on the high temperature corrosion of flame sprayed power boiler components with nickel alloys powders. The experiment results show that coatings made of Ni-Cr-B-Si alloys show about 50% higher resistance to high-temperature corrosion than coatings made of Ni-B-Si alloys and much lower susceptibility to corrosion than low-alloy steel intended for boiler structures. The results of this manuscript are reasonably interesting and novelty is easily identified. Therefore, the manuscript could be published. However, some improvement appears necessary for the publication. Some detailed comments are as follows.
(1) What is the phase composition of the spray coating?
(2) It is suggested to use parabolic equation and parabolic oxidation rate constant (kp) to study the oxidation kinetics.
(3) The “Results and Discussion” section ends somewhat abruptly. An added new paragraph should emphasize the novel findings from this study, compared to previous investigations of the spray coatings, comment about limitations of this study, and provide suggestions for future research.
(4) The conclusion should be concision.
Author Response
Dear Reviewer 3
Please see the attachment.
Yours sincerely,
Artur Czupryński

Reviewer 4 Report
The article aims to determine the high-temperature corrosion resistance of flame-sprayed coatings containing self-fluxing powders (Ni-Cr-B-Si alloys) and non-fluxing powders (Ni-B-Si alloys) on a boiler steel substrate. The manuscript is well-written and organized, although the subject is not quite novel. The authors should emphasize the originality of the study, compared to other references. The current form of the manuscript requires minor revisions before publication in the Materials journal.
Some questions and observations are summarized below:
1. Which is the role of tetrachloroethylene in the preparation step of the samples and why is this specific chemical chosen, since it is known to be harmful to humans and environment?
2. Table 6 – All spraying parameters are constant except for air pressure. Why is the air pressure value different for the Ni-B-Si alloys, compared to the self-fluxing Ni-C-B-Si?
3. The term ”welding” is mistakenly used a few times in the manuscript (line 219 – welding surface, line 345 – weld metal). Actually, the powders were used to manufacture a coating by a flame spraying technique, not a welding technology, for joining parts.
4. From the four types of Ni-based powders, only two are presented in the microscopic images (Figure 3). How about the other two specimens?
5. In Figure 10 a and b, the peaks should be labeled on the diffractogram, as for the other two samples, for an easier comprehension.
Author Response
Dear Reviewer 4
Please see the attachment.
Yours sincerely,
Artur Czupryński

Round 2
Reviewer 1 Report
.
Reviewer 2 Report
Most comments are addressed.